# Contractile and Mechanical Properties of Quadriceps Muscles Measured by the Method of Tensiomyography (TMG) in Professional Soccer Players: A Systematic Review, Meta-Analysis, and Meta-Regression

**DOI:** 10.3390/bioengineering11121295

**Published:** 2024-12-20

**Authors:** Daniel Fernández-Baeza, Germán Díaz-Ureña, Cristina González-Millán

**Affiliations:** Faculty of Health, Universidad Francisco de Vitoria, 28223 Madrid, Spain; german.diaz@ufv.es (G.D.-U.); 24crgonzalez9@gmail.com (C.G.-M.)

**Keywords:** tensiomyography, thigh, soccer, knee, quadriceps, rectus femoris

## Abstract

Tensiomyography (TMG) is a non-invasive tool used to assess contractile properties. This systematic review aimed to accomplish the following: (1) Analyze quadriceps TMG parameters in professional football players during the season and compare them with reference values. (2) Assess the differences in TMG parameters between quadriceps muscles. A PRISMA-guided search in PubMed, Web of Science, and Sport Discus (up to March 2024) identified 139 studies. Twelve in-season professional soccer players (20–29 years old) and quadriceps tensiomyography parameters were included (muscle displacement, delay time, and contraction time). All the studies were assessed using the Newcastle–Ottawa scale, scoring 7/9 to 8/9, indicating good quality. The findings of this study were that of the nine parameters analyzed, three variables were found to differ significantly. The weighted mean values were as follows: rectus femoris (contraction time 30.11 ms, muscle displacement 8.88 mL, delay time, 24.68 ms), vastus medialis (contraction time 25.29 ms, muscle displacement 7.45 mL, delay time, 21.27 ms), and vastus lateralis (contraction time 23.21 ms, muscle displacement 5.31 mL, delay time, 21.89 Â ms). Furthermore, significant differences were observed in muscle displacement between the rectus femoris and vastus medialis, and between the rectus femoris and vastus lateralis. The TMG can serve as a valuable device for assessing neuromuscular function in soccer players.

## 1. Introduction

Evaluating muscle function is essential for designing programs aimed at improving sports performance and preventing injuries [1]. In this context, tensiomyography (TMG) has emerged as a well-established, noninvasive, and objective method for assessing the mechanical and contractile properties of muscles over the past two decades [2,3]. TMG applies electrical stimulation to the superficial muscles, generating parameters that provide insights into muscle contraction time (Tc) measured in milliseconds, muscle displacement (Dm) measured in millimeters, which indicates muscle tone, and the following variables related to fatigue: relaxation time (Tr) measured in milliseconds, delay time (Td) measured in milliseconds, and maintenance time of contraction (Ts) measured in milliseconds [4,5]. Additionally, TMG provides data that enable the identification of lateral asymmetries (between both legs) and functional imbalances, such as those between agonist and antagonist muscles (e.g., hamstrings and quadriceps) [6].

The contraction time measures the time it takes for a muscle to reach its peak contraction after receiving an electrical or neural stimulus. It is an indicator of the muscle response speed. It has been observed that this parameter is a good indicator of the prevalence of fast-twitch fibers [7]. Shorter Tc is considered to reflect a higher rate of force production [8]. A player with a low Tc in the quadriceps has the advantages of powerful shots and acceleration. Conversely, a high Tc in certain stabilizing muscles may help sustain a prolonged effort.

Muscle displacement measures the extent to which a muscle deforms or displaces itself during contraction. It is considered to reflect muscle belly stiffness [9] and has been shown to be altered by muscle fatigue and ageing [8,10]. Dm and Tr have also been shown to increase with increasing muscle fatigue [11]. Different training states also reflect variations in TMG, with athletes with a prevalence of strength and power training having shorter Tc and smaller Dm [12,13,14]. Furthermore, it has been observed that Dm increases when atrophy occurs in the muscle architecture [15,16]. Players with an optimal Dm exhibit better muscle recovery and efficiency in repetitive actions such as running or passing. If a player shows low Dm in the leg muscles, they might experience issues such as early fatigue or limited mobility.

The delay time was defined as the interval between the stimulus and the start of the muscle contraction. This indicates neuromuscular efficiency and muscle readiness. A short Td is critical for actions such as sharp accelerations, quick directional changes, and fast reactions to dynamic game situations. A long Td can hinder a player’s ability to react swiftly, particularly under the physical and mental stress of an intense match. The relaxation time measures the time it takes for a muscle to return to its resting state after contraction. This reflects the muscle’s ability to recover and prepare for subsequent actions. In football, a short Tr in muscles such as the hamstrings or quadriceps allows players to perform repeated high-intensity efforts, such as sprints and tackles. A prolonged Tr can reduce effectiveness during prolonged play, increasing the risk of errors or injuries, particularly in the later stages of a match. The maintenance time of contraction measures how long a muscle can sustain contraction under load, reflecting muscular endurance and resistance to fatigue. Players with long Ts values in the core and leg muscles maintain strength and stability throughout the match, which is vital for midfielders and defenders. Short Ts may lead to reduced effectiveness in critical moments, such as holding defensive positions or sustaining pace during extended runs.

Muscle injuries are among the most prevalent types of injuries in football [17]. In total, 92% of all muscle injuries affect the four major muscle groups of the lower limbs: hamstrings (37%), adductors (23%), quadriceps (19%), and calf muscles (13%) [17]. Quadriceps injuries result in more missed games than hamstring and groin injuries, with a reinjury rate of 17% [17]. Quadriceps muscle strains are particularly common in sports involving repeated kicking and sprinting, making them a frequent occurrence in various forms of football worldwide [18].

Numerous studies have used TMG to assess soccer players [19,20,21,22,23,24,25,26]. However, it was not until this year that an interesting study conducted a systematic review of the literature to comprehensively synthesize existing studies reporting TMG-derived lower limb parameters in soccer players [27]. Regarding the use of TMG in injury prevention, to date, there is only one experimental study in which the number of hamstring injuries was reduced after implementing an individualized training program for each player-based TMG datum [20]. Another issue is that there is no consensus regarding appropriate values for professional football players. The TMG software provides specific reference values for all parameters, which represent the asverages of the values most frequently encountered in assessments performed by software developers (TMG-BMC Ltd., Ljubljana, Slovenia). Therefore, the aims of this study were twofold: (1) to analyze TMG parameters in the quadriceps of professional football players during the competitive season to determine the most frequent values and compare them with TMG software reference values. (2) To assess the differences in TMG parameters among the quadriceps muscles.

## 2. Materials and Methods

A systematic review was conducted from 10 November to 22 December 2023, to search for published scientific evidence on the contractile and mechanical properties of healthy professional soccer players during the competitive season. The reporting flow diagram of this systematic review was based on the Preferred Reporting Items for Systematic Reviews and Meta-Analyses (PRISMA) guidelines [28] (Figure 1). This review was registered in PROSPERO (CRD42024506270).

### 2.1. Search Strategy

The authors independently performed a literature search of PubMed, Web of Science, and Sport Discus electronic databases. In addition, the authors conducted a thorough search of the TMG-BMC website (TMG-BMC Ltd., Ljubljana, Slovenia, 2022) to identify additional articles. The following terms and their combinations were used as a search strategy string: (a) target population: “soccer” or “football” and “player,” or “; (b) intervention: “tensiomyography,” or “TMG”; (c) outcome corresponded to tensiomyography assessment. The inclusion criteria consisted of samples of professional male soccer players measured during the competitive season. The authors consulted other experts in the field to identify additional published studies.

### 2.2. Eligibility Criteria

Eligible studies were selected based on the criteria of healthy soccer or football players and the use of TMG, and all types of studies were included in this review, except case studies and reviews. Overall, the selected studies focused on changes in TMG-derived parameters.

The inclusion criteria were male sex, age between 20 and 29 years, and professional soccer players measured during the competitive season. Younger and female football players were not included because of the significant differences in physical, physiological, and biomechanical characteristics between male and female football players [29]. The focus of this study was on homogenizing the sample to more precisely analyze the impact of these variables on male football players, including women, which could introduce biases that would compromise the internal validity of the findings.

Studies reporting inappropriate populations for this review were excluded. Studies on teenagers and players aged > 29 years were excluded.

### 2.3. Study Selection and Data Extraction

Titles and abstracts from electronic searches were independently screened by the authors. The authors (D-F and G-D) then checked the full texts of the selected articles to consider the fit with eligibility criteria. In case of any disagreement between the authors, a third reviewer (C-G) was consulted to make the final decision. The authors (D-F and G-D) extracted the data separately. In case of disagreement, the third author cross-examined doubtful data. The following data were extracted: authors, year of publication, study population (sample size and age), and TMG parameters of the biceps femoris and semitendinosus knee flexors. These two muscles provide a more specific and relevant assessment of the contractile and mechanical properties that directly affect athletic performance and injury prevention. Three TMG variables were extracted: contraction time (Tc), which is fundamental for analyzing how quickly a muscle can generate force and directly influence its performance. Delay time (Td) is crucial for understanding muscle activation speed, which is essential in sports in which reaction time is key. Muscle displacement is a key indicator of muscle contraction and recovery capacity and is relevant for monitoring muscle fatigue and assessing the risk of injury. These three parameters (Td, Tc, and Dm) allow for a comprehensive evaluation of muscle performance and injury prevention, aligned directly with the objectives of the study in elite athletes. The relaxation time and sustain time variables were not extracted because, while complementary, they are not crucial for evaluating the central aspects of muscle performance that are of the greatest interest in this study. In case of missing data, the authors of the publications were contacted. We used the data for the dominant leg (kicking) for analysis. The TMG 3.6 software (TMG-BMC d.o.o., Ljubljana, Slovenia) calculated the average values with the specific muscle contractile and mechanical properties based on sex, age, and sport/position, regardless of laterality or dominant side. A weighted averaging technique was used to obtain the pooled averages and standard deviations of Tc, Dm, and Td.

### 2.4. Statistical Analyses

To enhance the reproducibility of the results, the weighted mean and standard deviation were calculated using the sample size for each study. Specifically, the mean value and standard deviation of the control group were derived from the weighted mean value and standard deviation based on the sample size of the three subgroups using the TMG software. The standard deviation for the control group was provided by software developers at our request. The standardized mean difference was used as the outcome measure, with data fitted to a random-effects model to account for the variability between studies.

Heterogeneity analysis involved estimating tau^2^ using the restricted maximum-likelihood estimator. Additionally, the Q test and I^2^ statistics were reported. To assess potential outliers and influential studies, studentized residuals and Cook’s distances were applied within the context of the model. Funnel plot asymmetry was examined using the rank correlation test (Begg and Mazumdar correlation test) and regression test (Egger regression), using the standard error of the observed outcomes as a predictor.

Cohen’s kappa (κ) was calculated to measure the inter-rater reliability for the two authors who selected the studies. The normalized reaction velocity (Vrn) was calculated using the formula 0.8/Tc [19]. To minimize the risk of overfitting, meta-regression was performed using age as a covariate. A scatter plot was created to display the original data points.

A linear mixed model (LMM) was used to compare the differences among muscles, with the sample size included as a random effect. To further investigate the differences between muscle groups, multiple comparisons were performed using Tukey’s test. Statistical analyses were conducted using the RStudio v. 2023.09.1+494, the Meta package v.6.5, the lme4 package v.1.1-35.5, and the ggplot2 package v.3.5.1 for graphical results.

### 2.5. Methodological Quality Assessment

The quality of the selected studies was assessed using the Newcastle–Ottawa scale (NOS). Articles with an NOS score of less than 7 were excluded. Subsequently, the risk of bias of the selected articles was reviewed by the authors (D-F and G-D). Any discrepancies between the authors were resolved through consultation with a third reviewer (C-G) for the final decision.

## 3. Results

The NOS score [30] varied from 7 to 8 in all studies, suggesting that all the articles were of good quality.

Table 1 lists the results for all combinations used in this review.

A flowchart of the study selection process is presented in Figure 1.

Table 2 shows the Egger regression, Begg and Mazumdar correlation tests, and Rosenthal Fail-Safe Number. The results indicated no substantial publication bias for most TMG variables across the muscles, as shown by the non-significant Egger regression and Begg–Mazumdar correlation tests. However, the significant R-F-S-N values suggest strong effect sizes for all the variables. The significant Begg–Mazumdar correlation for Dm in the VM muscle indicated a potential publication bias for this specific variable.

Table 3 presents the heterogeneity statistics of each variable. The results indicated significant heterogeneity across all the TMG variables and muscles, as evidenced by the high I^2^ and H^2^ values and significant Q statistics (*p*-values < 0.001).

Table 4 shows the values of the random effects model. Regarding the average outcome $\left(\hat{\mu }-value\right)$, in VM and VL, most of the variables were negative values. However, in the RF group, most were positive. The higher the positive values $\left(\hat{\mu }-value\right)$, the higher the values in the studies related to TMG. In the VM, both the Dm and Td showed significant differences. In the VL, only Tc showed significant differences. No significant differences were observed in the RF.

Table 5 shows the values of the meta-regression mixed-effects model using age as a covariable. No significant effects were observed, except for Tc in RF and VL. Although Tc in VM has no significant effect (*p* = 0.06), it has been considered. Figure 2 shows the results of the meta-regression analysis of the three variables. Each point represents the number of participants in a group. To illustrate the predictions of the REML model, a regression line was added to show the expected response time based on age. The confidence interval is based on the estimated variance of the predictions derived from the REML model.

Figure 3, Figure 4 and Figure 5 show the forest plot of the meta-regression results, including the model fitting weights, standardized mean differences (SMD), 95% confidence intervals (CI), and effect sizes for each included study. Each horizontal line represents the confidence interval for a specific study, and the size of the square indicates the weight of the study in the meta-analysis. The central vertical line represents the null effect (standardized mean difference (SMD) = 0). Negative values located on the central vertical line in Figure 3, Figure 4 and Figure 5 indicate lower values than those indicated by the TMG as normal values, whereas positive values located to the right of the central line indicate higher values than those indicated by the TMG as normal values. In Dm VM, most of the values included in the studies were lower than the TMG values. In addition, as shown in Table 4, this model was significantly different from the TMG values. All the included studies had similar weights in the fitting model. For both Td VM and Tc VL, most of the included studies were estimated to be positive, but one study was negative [31].

According to the studentized residuals and Cook’s distances, although some of the studies showed high values in both analyses due to sample size, the number of studies included in each variable, and their weights in the model fitting, none of the studies were excluded and considered to be overly influential on any of the variables. 

Table 5 shows the descriptive values of the muscles measured in each study. All the measurements were performed on the dominant leg. The authors who selected the studies to be included in this meta-analysis had an excellent inter-rater reliability (κ = 1). 

Table 6 shows the studies selected in this review, the study code to identify it in the forest plot, and the TMG parameter data for each muscle.

Each group in each study was considered an individual group to compare their values with the TMG values. The weighted average values are listed in Table 7.

Table 8 shows the values of the multiple comparisons related to the differences among the VL, VM, and RF.

There were significant differences between all the comparisons for DM. Regarding Td, there were significant differences in both VM–RF and VM–VL. No significant differences were observed in Tc.

## 4. Discussion

The objectives of this review were twofold: (1) to examine scientific evidence on TMG parameters in the quadriceps muscles of professional soccer players during the competitive season, comparing these findings with reference values provided by the software, and (2) to assess whether differences exist in the TMG parameters across different quadriceps muscles. Additionally, the most frequently recorded values represent the most repeated measurements of the contractile and mechanical properties in these athletes; however, the most frequently repeated values are not necessarily the optimal ones.

(1)Evidence on TMG parameters

The main findings of this study reveal that of the nine parameters analyzed, three variables differed significantly from the most frequently reported values by the TMG software, and from the scientific literature reviewed. Significant differences were identified in the vastus medialis for two variables. The first was Td (*p* < 0.001), a variable associated with fatigue [36]. The studies reported a lower Td than the software (21.27 ms vs. 22.90 ms, respectively) (Table 7), due to the players’ skill level. The software data included well-trained soccer players, whereas the participants in the studies were professional players. In elite players, fatigue induced by a soccer match appears less pronounced and recovery is more rapid than in lower-level players [37].

The other variable was muscle displacement (Dm) (*p* < 0.01), with lower Dm values in the studies (7.45 mm) than in the software (8.12 mm). This difference may also reflect discrepancies in the participant levels. Previous research has observed that playing level influences VM Dm, with elite players showing lower Dm (7.26 mm) than highly trained players (7.94 mm) [27].

Another significant difference was found in the contraction time (Tc) of the vastus lateralis (VL) (*p* < 0.04), with a higher Tc reported in the literature (23.21 ms) than in the software data (22.67 ms). This may be attributed to the timing of the measurements in the articles, which were taken during the competitive season, while the software measurements were taken during the preseason. It is well known that the cumulative load from training and matches can lead to fatigue, which is reflected in prolonged contraction time [11,36].

However, no significant differences were observed in any rectus femoris (RF) parameters (Table 4) between the TMG software data and the studies analyzed. This finding indicates that both the subjects in the studies and those in the software exhibited similar values because of the inherent physical demands of soccer, which shape the RF characteristics.

Age has a significant influence on tensiomyography (TMG)-derived parameters, reflecting broader trends in muscle mechanics and functionality. The meta-regression analysis revealed that the contraction time (Tc) increased with age across the quadriceps muscles, notably in the rectus femoris (RF) and vastus lateralis (VL), suggesting slower muscle responses, probably due to age-associated declines in the properties of type II (fast-twitch) muscle fibers and reductions in muscle elasticity. Although the changes in Tc for the vastus medialis (VM) did not reach statistical significance (*p* = 0.06), the confidence interval (−0.03 to 0.84) indicated a high probability (96.6%) that age-related modulation also affects this muscle. This suggests that contraction time increases with age, which is consistent with the findings of Paravlic [27]. These findings align with established evidence of age-related neuromuscular changes, including reduced activation efficiency, decreased mitochondrial density, increased muscle stiffness, and sarcopenia. This highlights the need to integrate age-specific approaches into training and rehabilitation programs to address the decline in muscle performance and mitigate the associated risk of injury.

(2)Differences across quadriceps muscles

When analyzing the differences between the rectus femoris (RF) and both vastus muscles, significant differences were observed in Dm between the RF and vastus medialis (VM) (*p* < 0.0351) and between the RF and vastus lateralis (VL) (*p* < 0.001). Dm was greater in the RF (8.88 mm) than in the VM (7.45 mm) and VL (5.31 mm), indicating that the rectus femoris exhibits a lower muscle tone. Dm is a parameter that reflects muscle tone and stiffness. A large displacement in the stimulated muscle suggests reduced muscle tone, whereas minimal displacement implies increased stiffness. Conversely, an optimal Dm level (depending on age, player level, and sport characteristics, among other variables) reflects adequate muscle tone [23]. In terms of contraction time measured in milliseconds in our study, the TMG measurements revealed that the contraction time (Tc) of the rectus femoris was slower (30.11 ms) than that of the vastus muscles, both the medialis (25.29 ms) and lateralis (23.21 ms). The contraction time reflects muscle contraction speed: a low contraction time indicates a high contraction speed, while a high contraction time suggests a slower muscle [38]. Thus, our findings suggest that the RF has a lower muscle tone and is slower than the vastus medialis and lateralis muscles. These differences in the quadriceps muscles were also noted in Paravlic’s 2024 study [27], specifically between the VL and RF, which found that the RF showed poorer Dm and Tc values than the VL at different competition levels.

These highly imbalanced TMG values between the RF and the vastus muscles could potentially increase the risk of injury to the rectus femoris. This could be because the rectus femoris is highly prone to injury due to its biarticular role as a knee extensor and hip flexor. Its dual-joint function is subject to combined stresses, especially during actions such as sprinting or kicking, when simultaneous hip flexion and knee extension increase the mechanical loads. Its susceptibility is heightened by its longer length and reduced cross-sectional area, limiting its ability to dissipate force. Additionally, overextension during extreme motion or fatigue-induced imbalances with other quadriceps muscles amplify the risk of strain, particularly at the musculotendinous junction. This is an important factor to be considered in injury prevention and rehabilitation.

It is important to note that the rectus femoris is the most frequently injured quadriceps muscle [39]. The rectus femoris extends the knee, flexes the hip, and stabilizes the pelvis over the femur during weight-bearing activities [40]. Muscle strain injuries typically occur during eccentric muscle actions. In soccer, both sprinting and kicking require eccentric action of the rectus femoris, and its biarticular nature contributes to vulnerability to injuries [18]. Furthermore, in Scurr et al.’s 2011 [41] study measuring electromyographic activity in the quadriceps group during the contact phase of a kick, the vastus lateralis was found to be the most active muscle during the contact phase (89%), followed by the vastus medialis (83%), and the rectus femoris as found to be the least active (77%). This imbalance among the quadriceps muscles, due to the biomechanical and anatomical characteristics of the RF, as well as its lower involvement in certain actions, may pose an injury risk.

The present review was limited by the number of studies analyzed. There were parameters that demonstrated significant differences between the software and the studies analyzed; however, the number of players measured with the TMG software was greater than that in the studies analyzed. To conclude, an additional limitation of the present study is the potential bias in the selection of the TMG parameters. The analysis prioritized Dm and Tc, as these are the most frequently reported parameters in the scientific literature, facilitating their interpretation and comparison with previous studies. However, this selection may have excluded relevant information provided by Tr and Ts, which are parameters that, although they are less studied and potentially more sensitive to fatigue or sensor positioning, offer a complementary perspective on muscle dynamics. This limitation could also have been influenced by the lower consistency with which the latter parameters were measured. Future research should consider a more comprehensive approach that incorporates all parameters to provide a more thorough assessment of muscle behavior. Finally, the reference data provided by the software did not differentiate between the playing levels of football players, such as professionals and amateurs.

## 5. Conclusions

The software values described the contractile properties of the quadriceps in football players, as there was no significant difference between the studies analyzed. The rectus femoris exhibited the lowest muscle tone and muscle strength compared to the other quadriceps muscles. Its values demonstrate a significant imbalance when compared to the vastus muscles, indicating vulnerability in its function and potential for injury. This imbalance may contribute to an increased risk during high-intensity movements, in which proper muscle coordination and strength are crucial for injury prevention. Addressing this discrepancy through targeted rehabilitation and training interventions could potentially improve muscle balance, enhance overall knee and hip function, and reduce the risk of strain and injury. Future research should focus on strategies to correct these imbalances and improve the performance of the rectus femoris considering its biarticular role in both knee extension and hip flexion.

## Figures and Tables

**Figure 1 bioengineering-11-01295-f001:**
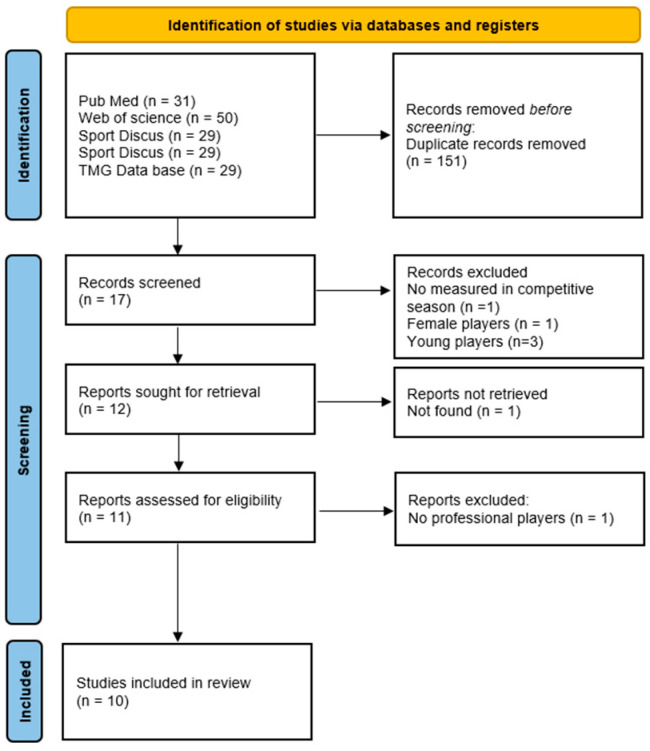
PRISMA flow diagram of the included studies.

**Figure 2 bioengineering-11-01295-f002:**
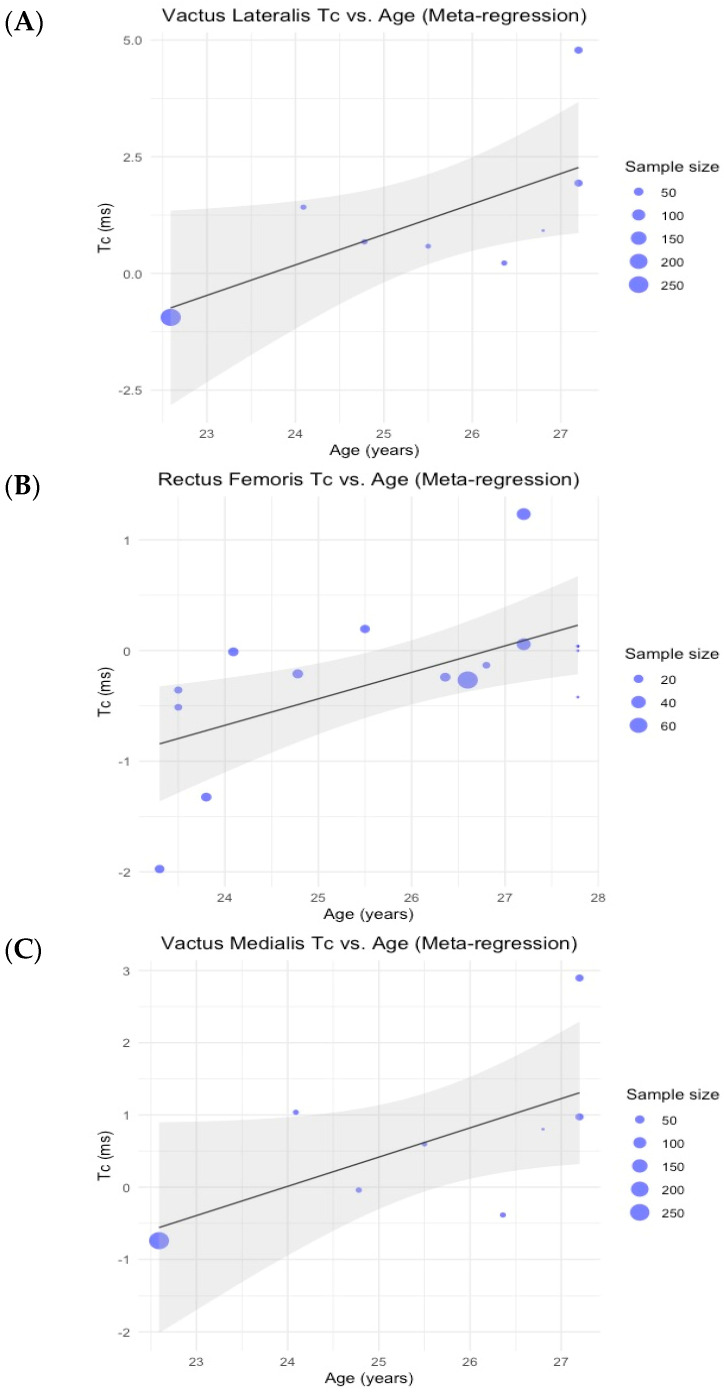
Results of the meta-regression analysis investigating the moderating effect of age on the time of contraction of the vastus lateralis (**A**), rectus femoris (**B**), and vastus medialis (**C**) muscles in soccer players. Blue dots represent primary studies, solid lines denote meta-regression prediction lines, and the gray area indicates the 95% confidence intervals around the mean.

**Figure 3 bioengineering-11-01295-f003:**
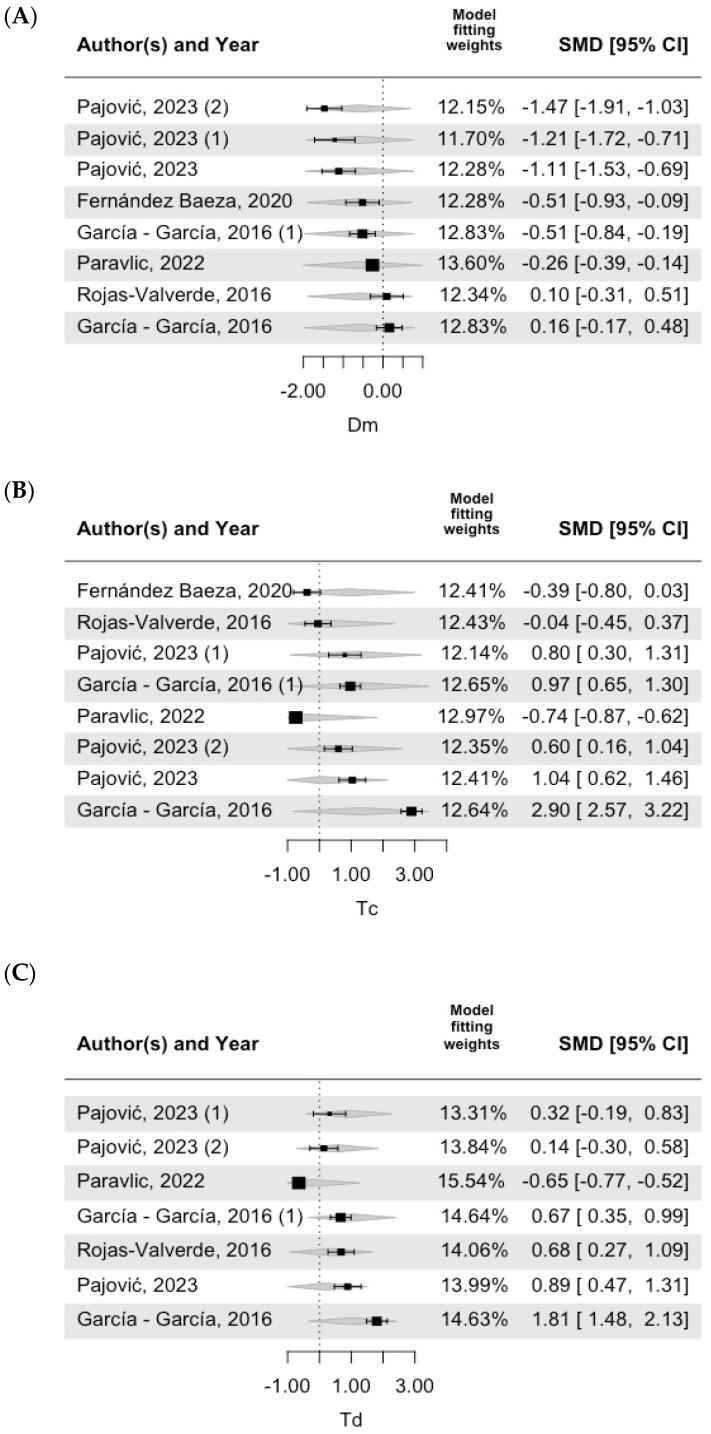
**VM forest plot.** Note: (**A**) Dm (muscle displacement), (**B**) Tc (contraction time), (**C**) Td (delay time) [26,32,33,34,35].

**Figure 4 bioengineering-11-01295-f004:**
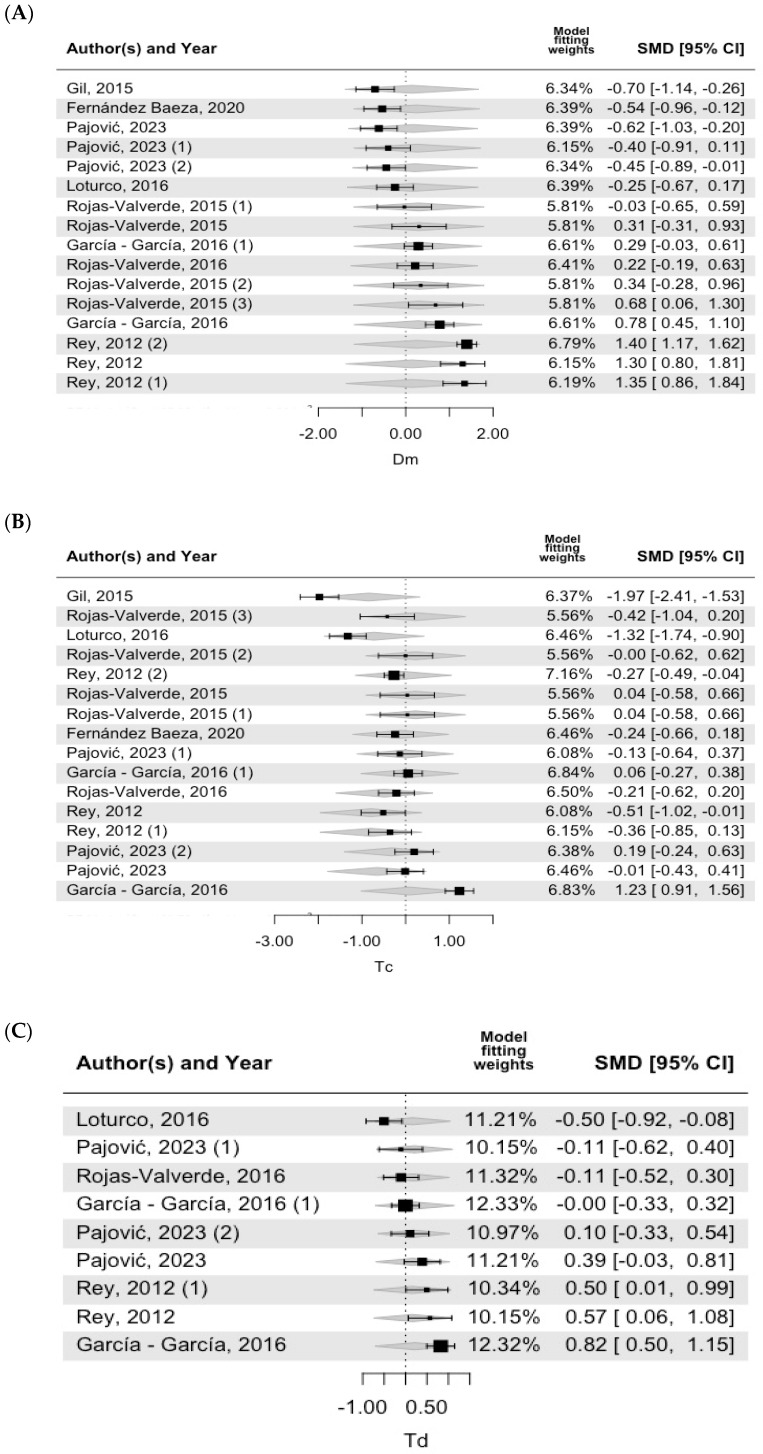
**RF forest plot.** Note: (**A**) Dm (muscle displacement), (**B**) Tc (contraction time), (**C**) Td (delay time) [19,20,22,23,24,26,33,34,35].

**Figure 5 bioengineering-11-01295-f005:**
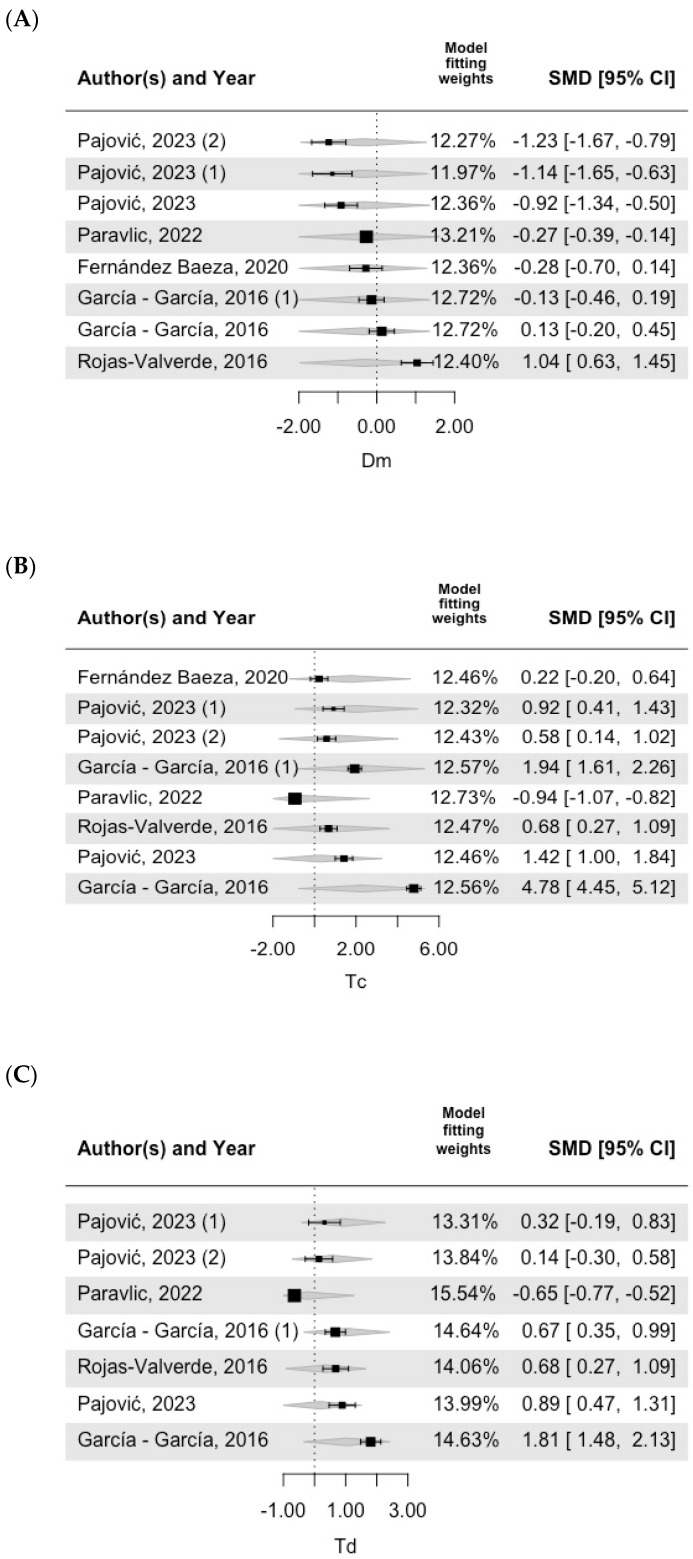
**VL forest plot.** Note: (**A**) Dm (muscle displacement), (**B**) Tc (contraction time), (**C**) Td (delay time) [26,32,33,34,35].

**Table 1 bioengineering-11-01295-t001:** Specific search equation results.

	PubMed	Sport Discus	Web of Science
Soccer and TMG	20	20	34
Soccer and tensiomyography	28	27	52
Football and TMG	8	12	16
Football and Tensiomyography	10	15	24
(((Soccer)) OR (football)) AND ((TMG)))	22	16	36
(((Soccer)) OR (football)) AND ((tensiomyography)))	31	16	55
(((Soccer players)) OR (football players)) AND ((tensiomyography)))	31	10	50
(((Soccer players)) OR (football players)) AND ((TMG)))	22	10	32
(((soccer players)) OR (football players)) AND ((TMG) OR (tensiomyography)))	31	29	50

**Table 2 bioengineering-11-01295-t002:** Egger regression, Begg-and-Mazumdar correlation test, and Rosenthal’s Fail-Safe Number.

Muscle	TMG Variable	Egger Regression Value (*p*-Value)	B-M Correlation Value (*p*-Value)	R-F-S-N Value (*p*-Value)
VM	Tc	0.541 (0.59)	0.07 (0.90)	142 (<0.001)
	Dm	−1.479 (0.14)	−0.64 (0.03)	219 (<0.001)
	Td	1.30 (0.19)	0.14 (0.77)	383 (<0.001)
RF	Tc	−0.332 (0.74)	−0.08 (0.69)	77 (<0.001)
	Dm	−0.44 (0.66)	0.00 (1)	147 (<0.001)
	Td	−0.20 (0.84)	0.16 (0.61)	16 (0.003)
	Tc	0.436 (0.66)	0.00 (1)	637 (<0.001)
VL	Dm	−0.783 (0.43)	−0.50 (0.11)	74 (<0.001)
	Td	0.887 (0.37)	−0.14 (0.77)	66 (<0.001)

Note: B-M correlation = Begg-and-Mazundar correlation test, R-F-S-N = Rosenthal’s Fail-Safe Number.

**Table 3 bioengineering-11-01295-t003:** Heterogeneity statistics.

Muscle	TMG Variable	I^2^	H^2^	Q	*p*-Value
VM	Tc	97.92	48.18	514.57	<0.001
	Dm	92.45	13.24	65.99	<0.001
	Td	83.77	6.16	28.54	<0.001
RF	Tc	90.62	10.66	175.48	<0.001
	Dm	91.17	11.32	210.361	<0.001
	Td	75.19	4.03	34.57	<0.001
	Tc	99.06	106.04	1193.98	<0.001
VL	Dm	95.07	20.29	84.44	<0.001
	Td	95.67	23.09	267.00	<0.001

**Table 4 bioengineering-11-01295-t004:** Random-effects model.

TMG Parameters	k	$\hat{\mu }$	CI (95%) Lower Bound	CI (95%) Upper Bound	Z	*p*
Tc VM	8	0.641	−0.145	1.426	1.60	0.11
Dm VM	8	−0.588	−1.003	−0.173	−2.77	<0.01 *
Td VM	7	−0.663	−0.965	−0.361	−4.30	<0.001 *
Tc RF	16	−0.241	−0.586	0.104	−1.37	0.17
Dm RF	16	0.232	−0.124	0.588	1.28	0.20
Td RF	9	0.186	−0.092	0.465	1.31	0.19
Tc VL	8	1.20	0.031	2.368	2.01	0.04 *
Dm VL	8	−0.342	−0.853	0.169	−1.31	0.19
Td VL	7	0.547	−0.025	1.119	1.87	0.06

Note: k = number of articles, $\hat{\mu }$ = Tau^2^ estimator: restricted maximum likelihood, CI = confidence interval, * = significant differences.

**Table 5 bioengineering-11-01295-t005:** Mixed-effects model.

TMG Parameters	k	$\hat{\mu }$	CI (95%) Lower Bound	CI (95%) Upper Bound	Z	*p*
Tc VM	8	0.405	−0.03	0.84	1.823	0.07
Dm VM	8	−0.002	−0.288	0.283	−0.017	0.99
Td VM	7	−0.093	−0.274	0.088	−1.001	0.31
Tc RF	16	0.239	0.068	0.411	2.734	<0.01
Dm RF	16	0.035	−0.181	0.252	0.32	0.75
Td RF	9	0.016	−0.183	0.214	0.155	0.88
Tc VL	8	0.652	0.031	1.274	2.057	0.04
Dm VL	8	−0.009	−0.363	0.345	−0.052	0.96
Td VL	7	0.279	−0.007	0.566	1.909	0.06

Note: k = number of articles, $\hat{\mu }$ = Tau^2^ estimator: restricted maximum likelihood, CI = confidence interval.

**Table 6 bioengineering-11-01295-t006:** Studies selected.

CODE	REFERENCES	AGE	SAMPLE	VM	RF	VL	TMG
Rey, 2012	Rey et al., 2012 [24]	23.5 (3.4)	15		28.2 (2.6)		Tc (ms)
					11.1 (3.9)		Dm (mm)
					25.7 (1.8)		Td (ms)
Rey, 2012 (1)	Rey et al., 2012 [24]	23.5 (3.4)	16		29.1 (4)		Tc (ms)
					11.2 (2.3)		Dm (mm)
					25.5 (1.8)		Td (ms)
Rey, 2012 (2)	Rey et al., 2012 [23]	26.6 (4.4)	78		29.63 (3.99)		Tc (ms)
					11.34 (3.09)		Dm (mm)
					25.91 (2.02)		Td (ms)
Loturco, 2016	Loturco et al., 2016 [19]	23.8 (4.2)	22		23.5 (3.9)		Tc (ms)
					7.28 (3.27)		Dm (mm)
					22.7 (2.7)		Td (ms)
Paravlic, 2022	Paravlic et al., 2022 [32]	22.59 (3.91)	57	22.92 (2.38)		19.91 (2.24)	Tc (ms)
				7.65 (1.32)		5.3 (1.43)	Dm (mm)
				21.21 (1.84)		20.63 (1.57)	Td (ms)
Fernández-Baeza 2020	Fernández et al., 2020 [33]	26.36 (4.68)	22	24.09 (2.57)	29.77 (9.5)	23.33 (3.07)	Tc (ms)
				7.2 (1.81)	6.56 (2.92)	5.28 (1.17)	Dm (mm)
Pajovic, 2023	Pajović et al., 2023 [26]	24.09 (5.1)	22	28.9 (6.3)	31.1 (6.2)	26.9 (5.76)	Tc (ms)
				6.13 (1.68)	6.37 (3)	4.31 (1.64)	Dm (mm)
				22.3 (1.46)	25.2 (4.66)	24.1 (3.72)	Td (ms)
Pajovic, 2023 (1)	Pajović et al., 2023 [26]	26.8 (3.7)	15	28.1 (6.03)	30.4 (5)	25.4 (5.71)	Tc (ms)
				5.95 (1.52)	6.9 (2.45)	3.97 (1.13)	Dm (mm)
				21.9 (1.56)	23.8 (1.91)	22.8 (1.83)	Td (ms)
Pajovic, 2023 (2)	Pajović et al., 2023 [26]	25.5 (5.7)	20	27.4 (4.15)	32.3 (7.92)	24.4 (3.67)	Tc (ms)
				5.48 (2.04)	6.79 (2.48)	3.83 (1.33)	Dm (mm)
				22.07 (1.66)	24.4 (2.77)	22.4 (1.39)	Td (ms)
Gil, 2015	Gil et al., 2015 [20]	23.3 (4.8)	20	19.74 (3.16)			Tc (ms)
				6.16 (1.9)			Dm (mm)
Rojas Valverde, 2015	Rojas-Valverde et al., 2015 [22]	27.78 (2.87)	10	31.39 (4.58)			Tc (ms)
				8.64 (2.63)			Dm (mm)
Rojas Valverde, 2015 (1)	Rojas-Valverde et al., 2015 [22]	27.78 (2.87)	10	31.4 (4.4)			Tc (ms)
				7.81 (2.06)			Dm (mm)
Rojas Valverde, 2015 (2)	Rojas Valverde et al., 2015 [22]	27.78 (2.87)	10	31.16 (4.49)			Tc (ms)
				8.73 (2.33)			Dm (mm)
Rojas Valverde, 2015 (3)	Rojas-Valverde et al., 2015 [22]	27.78 (2.87)	10	28.73 (4.28)			Tc (ms)
				9.57 (2.18)			Dm (mm)
García-García, 2016	García et al., 2016 [34]	27.2 (3.3)	37	35.2 (5.4)	38.3 (3.3)	36.9 (4.4)	Tc (ms)
				8.4 (1.4)	9.8 (2.4)	5.9 (1.5)	Dm (mm)
				20.7 (1.3)	26.4 (1.6)	26.2 (4)	Td (ms)
García-García, 2016 (1)	García et al., 2016 [34]	27.2 (3.3)	37	28.7 (6.7)	31.5 (5.8)	28.5 (7.2)	Tc (ms)
				7.2 (1.1)	8.6 (2.4)	5.5 (1.8)	Dm (mm)
				20.2 (1.4)	24.1 (2.1)	23.6 (3)	Td (ms)
Rojas-Valverde, 2016	Rojas-Valverde et al., 2016 [35]	24.78 (3.9)	23	(25.25) (2.95)	29.95 (2.32)	24.68 (4)	Tc (ms)
				8.29 (1.49)	8.42 (3.15)	7.29 (2.47)	Dm (mm)
				22.44 (1.56)	23,8 (1.27)	23.62 (1.38)	Td (ms)

**Table 7 bioengineering-11-01295-t007:** Weighted average values in the studies analyzed and the TMG software of TMG parameters.

	VM_RA	VM_SOFT	RF_RA	RF_SOFT	VL_RA	VL_SOFT
Tc (ms)	25.29 (3.43)	25.39 (3.37)	30.11 (4.66)	31.17 (5.80)	23.21 (3.33)	22.67 (2.96)
n	442	5120	367	5552	442	5117
Dm (mm)	7.45 (1.4)	8.12 (1.79)	8.88 (2.74)	7.89 (2.46)	5.31 (1.5)	5.71 (1.52)
n	442	5120	367	5552	442	5117
Td (ms)	21.27 (1.70)	22.90 (2.12)	24.68 (2.26)	24.11 (2.79)	21.89 (2.01)	22.08 (2.27)
n	420	5120	207	5552	420	5117
Vrn (mm/ms)	0.0322	0.0315	0.0271	0.0257	0.0358	0.0353
n	442	5120	367	5552	442	5117

Note: VM_RA = vastus medialis research article; VM _SOFT = vastus medialis software TMG; RF_RA = rectus femoris research article; RF _SOFT = rectus femoris software TMG; VL_RA = vastus lateralis research article; VL _SOFT = vastus lateralis software TMG.

**Table 8 bioengineering-11-01295-t008:** Multiple comparisons of means: Tukey’s contrasts.

TMG Variable	Parameter	Estimate	Std. Error	t Value/z Value	*p*-Value	95% CI (Lower)	95% CI (Upper)
Dm	VL-RF	−3.08	0.49	−6.29	<0.001	−4.22	−1.93
	VM-RF	−1.21	0.49	−2.48	0.035	−2.36	−0.07
	VM-VL	1.87	0.52	3.61	<0.001	0.65	3.08
Tc	VL-RF	−3.39	1.51	−2.24	0.065	−6.94	0.16
	VM-RF	−2.07	1.51	−1.37	0.358	−5.62	1.48
	VM-VL	1.32	1.64	0.81	0.700	−2.52	5.15
Td	VL-RF	−1.29	0.64	−2.00	0.112	−2.80	0.22
	VM-RF	−3.08	0.64	−4.78	<0.001	−4.59	−1.57
	VM-VL	−1.79	0.68	−2.62	0.024	−3.39	−0.19

Note: Std. Error = standard error, CI = confidence interval.

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
