# Peer review of "Contractile and Mechanical Properties of Quadriceps Muscles Measured by the Method of Tensiomyography (TMG) in Professional Soccer Players: A Systematic Review, Meta-Analysis, and Meta-Regression"

_bioengineering, 2024, doi:10.3390/bioengineering11121295_

Round 1

Reviewer 1 Report

Comments and Suggestions for Authors

The current systematic review aimed to examine tensiomyography (TMG), a non-invasive tool used to assess contractile properties during an isometric muscle contraction, as a tool for assessing neuromuscular function in professional football players’ quadriceps during the competitive season. Twelve studies (ages 20–29) were analyzed for contraction time, maximal displacement, and delay time across the rectus femoris, vastus medialis, and vastus lateralis.

Please find the suggestions below.

Abstract

1.     The subheadings like Objective:, Data source:, Study Selection:, Data extraction:, and so on should be removed.

2.     The abstract is lengthy. It should be concise and not exceed the journal guideline (<200 words).

3.     Please check the font and format of the abstract and the whole manuscript.

Introduction

1.     Please introduce objectives upfront, followed by a clear explanation of TMG principles, applications, and its relevance to injury prevention.

2.     Please explain more about how muscle contraction time (Tc), muscle displacement (Dm) can indicate muscle tone, and how relaxation time (Tr), delay time 42 (Td), and maintenance time of contraction (Ts) variables are related to fatigue.

3.     Please add the unit of each TMG’s parameters.

4.     Please provide more context for key findings, especially the relationship between TMG parameters and practical outcomes (e.g., reducing injuries), especially in soccer players.

Materials and methods

1.     Please discuss how excluding certain parameters is conducted and the study’s applicability, e.g., studies reporting an inappropriate population.

2.     Please describe how the control group’s mean and standard deviation were derived from the TMG software. Additional details on how these values were pooled or averaged would enhance reproducibility.

3.     The references are needed to be added in most sections, e.g., in the section 2.3 Study selection and data extraction.

4.     The reference of the Newcastle-Ottawa scale (NOS) should be added.

Results

1.     Figure 1 should be placed at the end of the first paragraph of Materials and Methods.

2.     Figures are small and less clear and detailed.

3.     In table 6, adding sub-columns of the MUSCLES MEASURED for each muscle and parameter would make the table clearer. Additionally, the word dominant leg in the table can be omitted or added in the note shown in the table caption to make the table clean.

4.     In Table 7, converting the table rows and columns would help the audience to easily compare between the review articles and TMC software.

Discussion

1.     Please provide more details on how age affects TMG parameters. Discuss whether age-related changes align with broader trends in muscle mechanics and fatigue resistance.

2.     Please extend the discussion of VM, VL, and RF injury risk to include how findings could inform training or rehabilitation strategies for soccer players.

3.     Please use clearer transitions between findings and implications to avoid abrupt topic shifts, particularly when moving between muscle groups. Maybe grouping the discussion based on the objectives would make it easy to follow.

4.     Please clarify the implications of marginally significant findings (e.g., p < 0.06) and whether they warrant further investigation or cautious interpretation.

5.     Please discuss potential limitations of the study.

Conclusion

1.     Please replace vague terms like "reality" with specific descriptions of how the TMG software values align with observed contractile properties in soccer players.

2.     Please define what "adequate values" mean in the context of contractile properties and explain how they are determined.

3.     In the conclusion, using the full term of the parameters is possible in order to help the audience to summarize the findings.

Comments on the Quality of English Language

Please check the format and some grammatical errors.

Author Response

Abstract:

  1. The subheadings like Objective:, Data source:, Study Selection:, Data extraction:, and so on should be removed.

Responde: Removed

  1. The abstract is lengthy. It should be concise and not exceed the journal guideline (<200 words).

Response: Reduced

  1. Please check the font and format of the abstract and the whole manuscript.

Response: Corrected

Introduction

  1. Please introduce objectives upfront, followed by a clear explanation of TMG principles, applications, and its relevance to injury prevention.

Response:

Response: In the introduction section “Regarding the use of TMG in injury prevention, to date, THERE IS ONLY ONE experimental study in which the number of ham-string injuries was reduced after implementing an individualized training program for each player based on TMG data [20].

  1. Please explain more about how muscle contraction time (Tc), muscle displacement (Dm) can indicate muscle tone, and how relaxation time (Tr), delay time 42 (Td), and maintenance time of contraction (Ts) variables are related to fatigue.

Response: Added more information

  1. Please add the unit of each TMG’s parameters.

Response: Added in the text

  1. Please provide more context for key findings, especially the relationship between TMG parameters and practical outcomes (e.g., reducing injuries), especially in soccer players.

Response: In the introduction section “Regarding the use of TMG in injury prevention, to date, THERE IS ONLY ONE experimental study in which the number of ham-string injuries was reduced after implementing an individualized training program for each player based on TMG data [20].

Materials and methods

  1. Please discuss how excluding certain parameters is conducted and the study’s applicability, e.g., studies reporting an inappropriate population.

Response: “Inclusion criteria were male, ages between 20 to 29 years, professional soccer player, measured in competitive season. Younger players and women football players were not included due to the significant differences in physical, physiological, and biomechanical characteristics between male and female football players. Since the focus of the study is to homogenize the sample to more precisely analyze the impact of these variables on male football players, including women could introduce biases that would compromise the internal validity of the findings”

  1. Please describe how the control group’s mean and standard deviation were derived from the TMG software. Additional details on how these values were pooled or averaged would enhance reproducibility.

              Reponse: I have rewritten the first paragraph of 2.4 Statistical analysis

  1. The references are needed to be added in most sections, e.g., in the section

2.3 Study selection and data extraction.

  1. The reference of the Newcastle-Ottawa scale (NOS) should be added.

Respone: Added

Results

  1. Figure 1 should be placed at the end of the first paragraph of Materials and Methods.

Response: we follow PRISMA guidelines the flow diagram have to be in the results

  1. Figures are small and less clear and detailed.

              Response: I have written new text starting at line 228 and onwards, and rewriteen the paragraph that starts at line 244

  1. In table 6, adding sub-columns of the MUSCLES MEASURED for each muscle and parameter would make the table clearer. Additionally, the word dominant leg in the table can be omitted or added in the note shown in the table caption to make the table clean.

  1. In Table 7, converting the table rows and columns would help the audience to easily compare between the review articles and TMC software.

              Response: Done

 Discussion

  1. Please provide more details on how age affects TMG parameters. Discuss whether age-related changes align with broader trends in muscle mechanics and fatigue resistance.

Response: Added more information Age exerts a significant influence on Tensiomyography (TMG) derived parameters, reflecting broader trends in muscle mechanics and functionality. Meta-regression analysis reveals that contraction time (Tc) increases with age across the quadriceps muscles, notably in the rectus femoris (RF) and vastus lateralis (VL), suggesting slower muscle responses likely due to age associated declines in the properties of type II (fast-twitch) muscle fibers and reductions in muscle elasticity. Although changes in Tc for the vastus medialis (VM) did not reach statistical significance (p = 0.06), the confi-dence interval (-0.03 to 0.84) indicates a high probability (96.6%) that age-related mod-ulation also affects this muscle. This suggests that contraction time increases with age, aligning with findings by Paravlic [27]. These findings align with established evidence of aging-related neuromuscular changes, including reduced activation efficiency, de-creased mitochondrial density, increased muscle stiffness, and sarcopenia. This high-lights the necessity of integrating age-specific approaches into training and rehabilitation programs to address the decline in muscle performance and mitigate the associated risk of injury. 

Response: regarding fatigue it is not a objective of the study.

  1. Please extend the discussion of VM, VL, and RF injury risk to include how findings could inform training or rehabilitation strategies for soccer players.

Response: Added more information. “These highly imbalanced TMG values between the RF and the Vastus muscles could potentially increase the risk of injury to the rectus femoris. This could be due to the rectus femoris is highly prone to injury due to its biarticular role as a knee extensor and hip flexor. Its dual-joint function subjects it to combined stresses, especially during ac-tions like sprinting or kicking, where simultaneous hip flexion and knee extension in-crease mechanical loads. Its susceptibility is heightened by a longer muscle length and reduced cross-sectional area, limiting its ability to dissipate force. Additionally, over-extension during extreme motion or fatigue-induced imbalances with other quadriceps muscles amplifies the risk of strain, particularly at the musculotendinous junction”.

  1. Please use clearer transitions between findings and implications to avoid abrupt topic shifts, particularly when moving between muscle groups. Maybe grouping the discussion based on the objectives would make it easy to follow.

Response: We have separated it by objectives and more paragraphs

  1. Please clarify the implications of marginally significant findings (e.g., p < 0.06) and whether they warrant further investigation or cautious interpretation.

              RESPONSE: I have added more information starting at line 325 and onwards. In addition, I have created a new figure added in figure 2

  1. Please discuss potential limitations of the study.

Response: The present review is limited by the number of studies analysed. There are pa-rameters that show significant differences between the software and the studies ana-lysed, but the number of players measured in the TMG software is higher than in the studies analysed. To conclude, another limitation of the present study lies in the po-tential bias in the selection of TMG parameters. The analysis prioritized Dm and Tc, as these are the most commonly reported parameters in the scientific literature, facilitat-ing their interpretation and comparison with previous studies. However, this choice may have excluded relevant information provided by Tr and Ts, parameters that, alt-hough less studied and potentially more sensitive to fatigue or sensor positioning, of-fer a complementary perspective on muscle dynamics. This limitation could also be in-fluenced by the lower consistency in measuring these latter parameters. Future re-search should consider a more inclusive approach that incorporates all parameters to provide a more comprehensive assessment of muscle behavior. Finally, the reference data provided by the software does not distinguish between the playing levels of foot-ball players, such as professionals and amateurs.

Conclusion

  1. Please replace vague terms like "reality" with specific descriptions of how the TMG software values align with observed contractile properties in soccer players.

Response: removed “reality”

  1. Please define what "adequate values" mean in the context of contractile properties and explain how they are determined.

Response: removed adequate values

  1. In the conclusion, using the full term of the parameters is possible in order to help the audience to summarize the findings.

              Response: “The software values describe the contractile properties of the quadriceps in football players, as there is no great difference with the studies analysed and the studies analized. The rectus femoris demonstrates the poorest muscular tone and the lowest muscle strength compared to other quadriceps muscles. Its values are significantly imbalanced when compared to the Vasti muscles, highlighting a vulnerability in its function and potential for injury. This imbalance may contribute to increased risk during high-intensity movements, where proper muscle coordination and strength are essential for injury prevention. Addressing this discrepancy through targeted rehabilitation and training interventions could improve muscle balance, enhance overall knee and hip function, and reduce the risk of strain and injury. Future studies should focus on strategies to correct these imbalances and improve the performance of the rectus fem-oris, considering its biarticular role in both knee extension and hip flexion.”

Reviewer 2 Report

Comments and Suggestions for Authors

Abstract

·      Line 11-14: Please specify what parameters will be taken into consideration.

·      The conclusions section contains the results, which should be moved earlier

·      Additionally, I cannot understand why these results are relevant. Please be clearer in the methods.

·      Last, please avoid repetitions (e.g., furthermore…)

Introduction

·      Please either use or not abbreviations. I strongly recommend avoiding any abbreviation since not necessary.

·      The first two paragraphs should be merged

·      Overall, the introduction should focus much more on the parameters rather than speculating on possible unproven and unrelated topics, e.g., injuries. Therefore, I strongly recommend revising the introduction to let me understand why the present study is necessary. As in the present form, the rationale is far from being clear.

Results,

·      The graphs are not clear. First, please remove any abbreviation and spell out what is needed, so I don’t have to search for it in the caption. Second, the X-axis of each graph needs more detail. Please revise the figures.

Discussion

The discussion mostly follows a speculative lane about the possible use of these parameters in practice, which I disagree with. Please stay on the results without repeating them, and let us understand what these could actually imply.

Author Response

REVIEWER 2

Abstract

  • Line 11-14: Please specify what parameters will be taken into consideration.

Response:  Added the TMG parameters used

  • The conclusions section contains the results, which should be moved earlier

Response: Moved earlier

  • Last, please avoid repetitions (e.g., furthermore…)

              Response: Avoided furthermore

Introduction

  • Please either use or not abbreviations. I strongly recommend avoiding any abbreviation since not necessary.

              Response: removed some abbreviations

  • The first two paragraphs should be merged

              Response: merged

  • Overall, the introduction should focus much more on the parameters rather than speculating on possible unproven and unrelated topics, e.g., injuries. Therefore, I strongly recommend revising the introduction to let me understand why the present study is necessary. As in the present form, the rationale is far from being clear.

Response: In the introduction section “Regarding the use of TMG in injury prevention, to date, THERE IS ONLY ONE experimental study in which the number of ham-string injuries was reduced after implementing an individualized training program for each player based on TMG data [20].

Results,

  • The graphs are not clear. First, please remove any abbreviation and spell out what is needed, so I don’t have to search for it in the caption. Second, the X-axis of each graph needs more detail. Please revise the figures.

              Response: Added a note in the figures explaining

Discussion

The discussion mostly follows a speculative lane about the possible use of these parameters in practice, which I disagree with. Please stay on the results without repeating them, and let us understand what these could actually imply.

Response: Added more information. “These highly imbalanced TMG values between the RF and the Vastus muscles could potentially increase the risk of injury to the rectus femoris. This could be due to the rectus femoris is highly prone to injury due to its biarticular role as a knee extensor and hip flexor. Its dual-joint function subjects it to combined stresses, especially during actions like sprinting or kicking, where simultaneous hip flexion and knee extension increase mechanical loads. Its susceptibility is heightened by a longer muscle length and reduced cross-sectional area, limiting its ability to dissipate force. Additionally, over-extension during extreme motion or fatigue-induced imbalances with other quadriceps muscles amplifies the risk of strain, particularly at the musculotendinous junction. This is an important factor to consider for injury prevention and rehabilitation”.

Round 2

Reviewer 1 Report

Comments and Suggestions for Authors

Thank you for the revised version of your manuscript. The revisions are noted, but upon further evaluation, several points require additional clarification or refinement to ensure the manuscript meets the journal's standards for scientific rigor and clarity:

  1. Grammatical errors and format checking are required.
  2. Adding the abbreviation of each parameter unit in the abstract would help the audience understand the meaning of the findings.
  3. Missing references, such as “female football players were not included due to the significant differences in physical, physiological, and biomechanical characteristics between male and female football players” (lines 134–135). Please check the entire manuscript.
  4. The quality of all the figures is low; upgrading the resolution and size is needed.
Comments on the Quality of English Language
    1. Grammatical errors and format checking are required.

Author Response

Thank you for the revised version of your manuscript. The revisions are noted, but upon further evaluation, several points require additional clarification or refinement to ensure the manuscript meets the journal's standards for scientific rigor and clarity:

  1. Comments 1: Grammatical errors and format checking are required.

Response 1: All authors have reviewed the manuscript to ensure grammatical and format accuracy.

  1. Comments 2: Adding the abbreviation of each parameter unit in the abstract would help the audience understand the meaning of the findings.

Response 2: added in the abstract “The weighted mean values found were rectus femoris (contraction time 30.11 ms, muscle displacement 8.88 ml, delay time 24.68 ms), vastus medialis (contraction time 25.29 ms, muscle displacement 7.45 ml, delay time 21.27 ms), and in vastus lateralis (contraction time 23.21 ms, muscle displacement 5.31 ml, delay time 21.89 ms).

  1. Comments 3: Missing references, such as “female football players were not included due to the significant differences in physical, physiological, and biomechanical characteristics between male and female football players” (lines 134–135). Please check the entire manuscript.

Response 3: Added a new reference (29) “Sex-Based Differences in Skeletal Muscle Kinetics and Fiber-Type Composition” by Haizlip et al. 2015

  1. Comments 4 The quality of all the figures is low; upgrading the resolution and size is needed.

Response 4: added new figures with a best resolution and we changed the size

Reviewer 2 Report

Comments and Suggestions for Authors

The authors have improved the manuscript. However, I question the real usefulness of the TMG and I am skeptical about any possible inference from it. Thus, on the one hand, I found the manuscript well written now; on the other hand, I disagree with the topic. 

In light of the above, I would recommend accepting the manuscript if the Editor disagrees with me and has a different vision. 

Author Response

Comments1: 

The authors have improved the manuscript. However, I question the real usefulness of the TMG and I am skeptical about any possible inference from it. Thus, on the one hand, I found the manuscript well written now; on the other hand, I disagree with the topic. 

In light of the above, I would recommend accepting the manuscript if the Editor disagrees with me and has a different vision.

Response 1: 

We appreciate your comments and understand your concerns regarding the usefulness of TMG. We would like to point out that TMG has more than 300 scientific publications supporting its validity and application in various fields. These publications demonstrate its usefulness and the inferences that can be drawn from its use. 

You can find the list of all scientific publications at this link: (https://www.tmg-bodyevolution.com/research/tmg-list-of-publications-3/)

We hope this information clarifies any doubts, and we are available to discuss any additional aspects you may consider necessary.